# The Fate of Sulfur Radical Cation of N-Acetyl-Methionine: Deprotonation vs. Decarboxylation

Katarzyna Grzyb [1], Vidhi Sehrawat [1] and Tomasz Pedzinski [1,2,*]

1    Faculty of Chemistry, Adam Mickiewicz University, 61-614 Poznan, Poland
2    Center for Advanced Technology, Adam Mickiewicz University, 61-614 Poznan, Poland
*    Correspondence: tomekp@amu.edu.pl

**Abstract:** In the present study, we investigated the photooxidation of the biomimetic model of C-terminal methionine, N-Acetyl-Methionine (N-Ac-Met), sensitized by a 3-Carboxybenzophenone (3CB) excited triplet in neutral and basic aqueous solutions. The short-lived transient species that formed in the reaction were identified and quantified by laser flash photolysis and the final stable products were analyzed using liquid chromatography coupled with high-resolution mass spectrometry (LC-MS) and tandem mass spectrometry (MSMS). Based on these complementary methods, it was possible to calculate the quantum yields of both competing reactions, and the deprotonation was found to be favored over decarboxylation (for neutral pH: $\phi_{-H} = 0.23$ vs. $\phi_{-CO2} = 0.09$, for basic pH: $\phi_{-H} = 0.23$ vs. $\phi_{-CO2} = 0.05$). Findings on such a model system, which can possibly mimic the complex protein environment, are important in understanding complicated biological systems, for example, the studied compound, N-Ac-Met, can, to some extent, mimic the methionine in the C-terminal domain of β-amyloid, which is thought to be connected with the pathogenesis of Alzheimer's disease.

**Keywords:** methionine oxidation; radical coupling; decarboxylation





## 1. Introduction

Methionine (Met) is a hydrophobic amino acid with an oxidatively labile thioether group. The oxidative modifications of methionine residue (resulting from the free radical reactions involving neighboring groups) acids are partially reversible via a complex process of enzymatic reduction, playing an important role in the maintenance and protection of the redox status [1].

Methionine in proteins is easily oxidizable, leading to an increase in the side-chain polarity of the residue [2]. As a result, its oxidation has long been regarded as a form of protein damage that occurs randomly in response to oxidative stress. For example, sulfur radical cation sites can interact with tertiary structures of the biosystems [1]. The oxidation also leads to the biological inactivation of hormones, changes in Met-containing proteins, and cataract formation in the eye lens [3]. The one-electron oxidation of methionine-containing compounds has also been closely linked to biological aging and the pathogenesis of neurodegenerative diseases, such as Alzheimer's disease, which are thought to be caused by oxidative stress and characterized by the pathological deposition of amyloid plaques in and around the brain tissue [4–6]. The primary component of the amyloid plaque is a β-amyloid (Aβ), which is a peptide generally composed of 40 or 42 amino acids containing methionine residue in position 35 (Met35) in its C-terminal domain. The presence of easily oxidized methionine plays a critical role in aggregation, the neurotoxicity of the Aβ, and secondary radical generation [6,7]. The exact role of Met in proteins is, however, still not fully understood [7]. There is also limited information regarding the possible photoproducts in hypoxic or anaerobic conditions, despite their occurrence in tumors and their destruction using cancer photodynamic therapies [8].

The strongly oxidating hydroxyl radicals ($\bullet$OH) and excited triplet states of benzophenones as one-electron oxidants have been widely used to study the primary steps of the oxidation of Met-containing peptides [9–15]. In the sensitized photooxidation of methionine-containing compounds, the excited triplet of a sensitizer (for example 3-Carboxybenzophenone, 3CB) accepts an electron from the sulfur moiety, resulting in a charge-transfer (CT) complex that can undergo the following processes: (i) back electron transfer ($k_{bet}$) yielding reagents in their ground states; (ii) charge separation ($k_{sep}$) yielding $3CB^{\bullet-}$ and sulfur radical cation ($>S^{\bullet+}$); (iii) in-cage proton transfer ($k_H$) from sulfur moiety to $3CB^{\bullet-}$ yielding relatively stable carbon-centered $\alpha$-thioalkyl radical ($\alpha$S). The sulfur radical cation as a primary oxidation product ($>S^{\bullet+}$) can, furthermore, be stabilized by the interaction with electron-rich atoms (S, N, O) resulting in the formation of inter- and intramolecular three-electron bonded species [16–18].

The fate of the sulfur radical cation is affected by the neighboring groups, for example, the presence of an amide bond eliminates the possibility of proton transfer from the protonated N-terminal group [19–21], and blocking the carboxyl group (for example by esterification) eliminates the possibility of decarboxylation [22,23]. The presence of neighboring amino acid residues with lone pairs from the carboxyl (Asp, Glu), amine (Lys), and hydroxyl groups (Thr, Ser) also impacts the reactions of the sulfur radical cation [24].

In this study, we investigated the one-electron 3CB-sensitized oxidation of the simplest model of C-terminal methionine—N-Acetyl-Methionine (N-Ac-Met, Figure 1). We focused on the competing reactions of sulfur radical cation: decarboxylation and deprotonation that could potentially occur in the methionine residue in the C-terminal domain of β-amyloid.

**Figure 1.** Structure of N-Acetyl-Methionine (N-Ac-Met) and the sensitizer 3-Carboxybenzophenone (3CB).

The stable products of the 3CB-sensitized oxidation of N-Ac-Met have been reported before. In this report, the oxidation products are studied by means of high-resolution mass spectrometry, coupled with high-performance liquid chromatography, steady-state photolysis, and time-resolved flash photolysis.

## 2. Experimental Section

### 2.1. Laser Flash Photolysis (LFP)

Nanosecond laser flash photolysis experiments were performed using 3-Carboxybenzophenone (3CB) as a photosensitizer. Samples containing 3CB and N-Ac-Met were excited using 355 nm, the third harmonic of an Nd: YAG laser (Spectral Physics Mountain View, CA, USA, model INDI 40-10) with pulses of 6–8 ns duration. The monitoring system consisted of a 150 W pulsed Xe lamp with a lamp Pulser (Applied Photophysics, Surrey, UK), a monochromator (Princeton Instruments, model Spectra Pro SP-2357, Acton, MA, USA), and an R955 model photomultiplier (Hamamatsu, Japan), powered by a PS-310 power supply (Stanford Research Systems, Sunnyvale, CA, USA). Nd: YAG laser acted as a pump and pulsed Xenon lamp acted as a probe of the exciting sample. A self-constructed flow system was used to avoid the depletion of the starting materials and the accumulation of possible photolysis products. The continuous circulation of aqueous solutions was achieved using a peristaltic pump, followed by simultaneous bubbling with a high purity Argon

for about an hour. All of the flash photolysis experiments were carried out in $1 \times 1$ cm rectangular quartz fluorescence cells. Kinetic traces were taken between 370 and 700, at 10 nm intervals. The time-resolved absorption spectra were constructed from the kinetic traces. For the determination of the quantum yields of the transients, relative actinometry was used, taking 3CB in an aqueous solution as the actinometer and $\varepsilon520 = 5400$ $M^{-1}$ $cm^{-1}$ for its triplet-triplet absorption. The concentration profiles were obtained via a multiple linear regression-based software (Decom) using the reference spectra data from the pulse radiolysis studies.

### 2.2. Steady-State Photolysis

Steady-state photolysis experiments were performed in a $1 \times 1$ cm rectangular cell on an optical bench irradiation system using a Genesis CX355STM OPSL laser from Coherent (Santa Clara, CA, USA), with 355 nm emission wavelength (the output power used was set at 50 mW). The concentrations of 3CB and N-Ac-Met were 4.5 mM and 20 mM, respectively.

### 2.3. Chemicals and Sample Preparation

3CB and N-Ac-Met were purchased from Sigma-Aldrich (St. Louis, MO, USA) and used as received. Water was purified through a Millipore (Merck Milli-Q) system. The pH of the solutions was adjusted by adding potassium hydroxide and/or hydrochloric acid, using a Mettler Toledo Five Easy FE20 pH-meter equipped with a semimicro InLab electrode from Mettler Toledo (Columbus, OH, USA). High purity argon (Linde) was used to purge the freshly prepared solutions of the reagents for photolysis experiments.

### 2.4. High Performance Liquid Chromatography (HPLC)

The HPLC system (Ultimate 3000, Thermo/Dionex) was equipped with an autosampler, a vacuum degasser, and a diode array detector. Solutions were injected without separation using Chromeleon 7.2. Two eluents were used for the separation and isolation of substrates and stable products after irradiation at 10 and 30 min: eluent A ($H_2O$) and eluent B ($CH_3CN$) with 0.1% (*v/v*) formic acid. The separation was achieved with a gradient from 7% to 60% of acetonitrile and water (with 0.1% formic acid), at a flow rate of 0.3 mL/min for 30 min, and the column temperature was set to 45 °C using a C18 reversed-phase analytical column (2.6 $\mu$m, 2.1 mm $\times$ 100 mm, Thermo-Scientific).

### 2.5. Liquid Chromatography-Mass Spectrometry (LC-MS)

The mass spectra were recorded using a hybrid time-of-flight mass spectrometer—QTOF (Impact HD, Bruker). Ions were generated by electrospray ionization (ESI) source under the following conditions: a flow rate of 0.3 mL/min, a nebulizer pressure of 1.5 bar, a capillary voltage of 4000 V, and a drying gas temperature of 200 °C.

## 3. Results

### 3.1. Laser Flash Photolysis

The transient absorption spectra obtained from the photosensitized oxidation of **1** (N-Ac-Met) were recorded after different times, after 355 nm laser pulse, and then deconvoluted into individual components (example of the spectral resolution for the time delay of 800 ns for pH 6.7 and 10.7 are presented in Figure 2, panels A and C). The resulting concentrations of each transient species were plotted as a function of time giving the concentration profiles (see Figure 2, panels B and D).

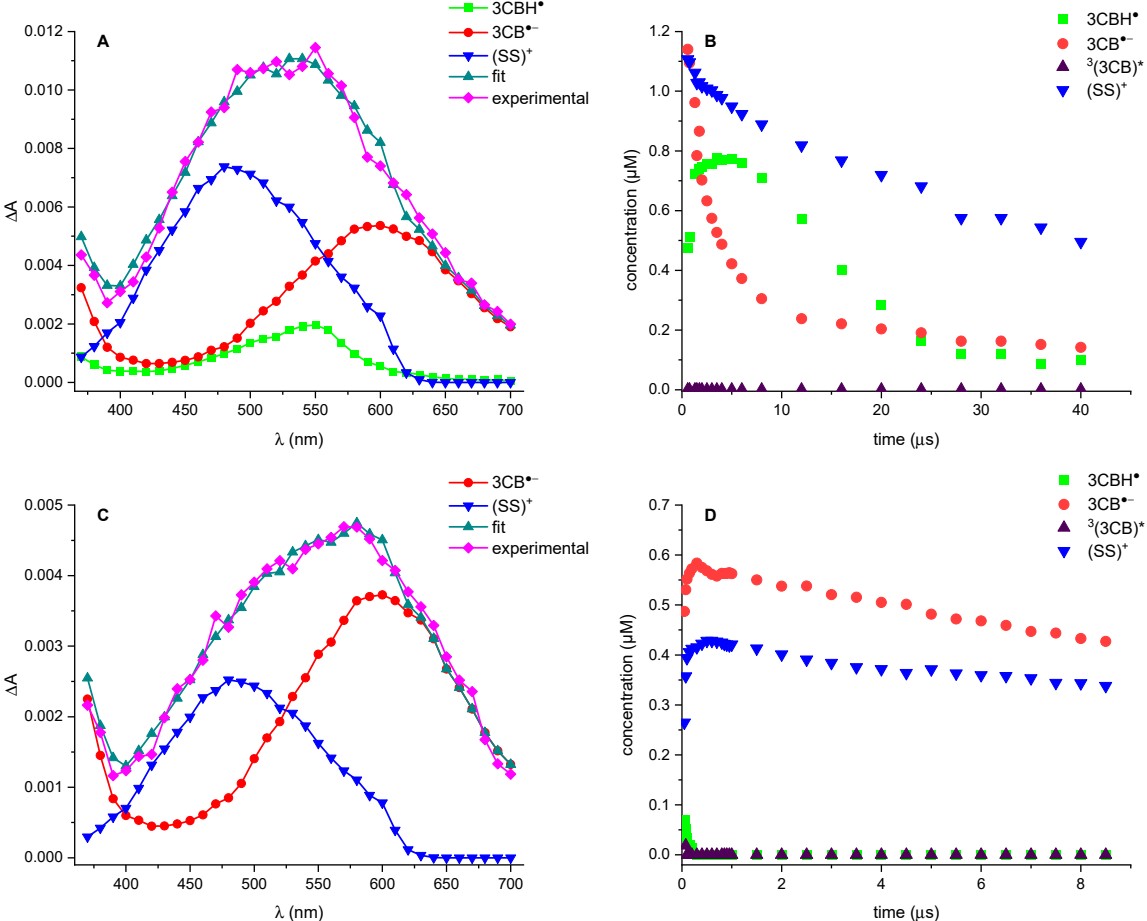

**Figure 2.** (**A**) Resolution of the spectral components in the transient absorption spectra 800 ns after laser pulse following quenching of the 3CB (4.5 mM) triplet state by N-Ac-Met (20 mM) at pH 6.7; (**B**) Concentration profiles calculated at different delay times with respect to the laser pulse for the reaction of 3CB (4.5 mM) excited triplet quenched by N-Ac-Met (20 mM) in aqueous solution at pH = 6.7; (**C**) Resolution of the spectral components in the transient absorption spectra 800 ns after laser pulse following quenching of the 3CB (4.5 mM) triplet state by N-Ac-Met (20 mM) at pH 10.7; (**D**) Concentration profiles calculated at different delay times with respect to the laser pulse for the reaction of 3CB (4.5 mM) excited triplet quenched by N-Ac-Met (20 mM) in aqueous solution at pH 10.7.

The same transient species are present at both pHs: ketyl radical 3CBH$^\bullet$, radical anion 3CB$^{\bullet-}$ and (S∴S)$^+$. The absence of the triplet excited state ($^3$(3CB)*), even at a shorter time, and the high concentration of (S∴S)$^+$ can be explained by the high concentration of the quencher (20 mM).

At pH 6.7, the initially formed 3CB$^{\bullet-}$ decays, while the concentration of 3CBH$^\bullet$ grows and then decreases, moving toward the equilibrium. On the other hand, in the basic solution, the dominant intermediate is 3CB$^{\bullet-}$; 3CBH$^\bullet$ is present only in small concentrations and for a very short time, and quickly decays.

No stabilization of N-Ac-Met through the formation of (S∴N)$^+$ was observed ($\lambda_{max}$ = 390 nm, $\varepsilon_{390nm}$ = 4520 M$^{-1}$ cm$^{-1}$) and no αN (α-amidoalkyl radical) was detected ($\lambda_{max}$ = 370 nm, $\varepsilon_{370nm}$ = 2000 M$^{-1}$ cm$^{-1}$ [25]). αS ((α-(alkylthio)alkyl radical) was not observed due to its absorption in the UV region $\lambda_{max}$ = 290 nm ($\varepsilon_{290nm}$ = 3000 M$^{-1}$ cm$^{-1}$) [17], below the experimentally available spectral region (overwhelmed by ground state absorption of 3CB).

The quantum yields, shown in Table 1, were calculated from the initial concentration of the intermediates, which are shown on the concentration profiles in Figure 2. 3CB in an

aqueous solution was used as an internal actinometer. The initial concentration of 3CB*
was calculated to be 3.46 µM (pH 6.7) and 1.70 µM (pH 10.7).

**Table 1.** Quantum Yields of Radical Species Generation from LFP Experiments.

| Radical Species | Φ (pH = 6.7) | Φ (pH = 10.7) |
|---|---|---|
| 3CB$^{\bullet-}$ | 0.32 | 0.28 |
| 3CBH$^{\bullet}$ | 0.20 | 0.10 |

### 3.2. LC-MS/MS

The Ar-saturated aqueous solutions containing sensitizer 3CB and N-Ac-Met at a
neutral pH were irradiated using a 355 nm continuous-wave laser. Before irradiation, only
the starting materials (3CB and N-Ac-Met) were detected using the LC-MS technique. The
irradiated samples at pH 6.7 and 10.7 were subjected to further LC-MS analysis.

The irradiation of the samples at both pHs yielded the same products. A representative
LC-MS analysis of the solution after 10 min irradiation at pH 6.4 is shown in Figure 3 (LC-
MS analysis of the sample before irradiation and irradiated for 30 min as well as the
chromatograms of the sample at pH 10.7 can be found in SI in Figure S1 and S2, see
Supplementary Materials).

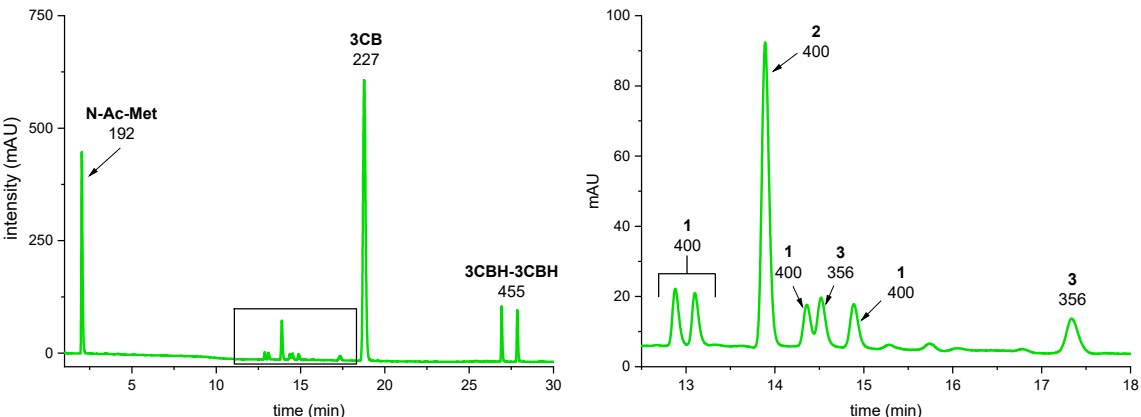

**Figure 3.** HPLC chromatogram of aqueous solutions containing: N-Ac-Met (20 mM) and 3CB
(4.5 mM) at pH 6.4 after 10-min irradiation. The expansion of the chromatograms between 12 and
18 min is shown next to the chromatogram. Peaks are labelled with m/z of the [M + H]$^{+}$ or [M +
H-H$_2$O]$^{+}$ ions of the acquired MS spectrum.

The peaks with m/z 192.0695 and 227.0705 were assigned to the substrates, N-Ac-Met,
and 3CB, respectively. The peaks with m/z 455.1501 and 455.1498 were assigned to the
products derived from the photosensitizer 3CB: dimers of 3CBH$^{\bullet}$ radicals [26]. Products **1**,
**2,** and **3** were formed in the photooxidation of N-Ac-Met and will be described more closely.

The accurate masses of products **1** and **2** are the same: m/z 400.1240, but their MSMS
fragmentations differ significantly, clearly suggesting their isomeric nature. Based on
the accurate masses and MSMS fragmentation pattern, it can be deduced that **1** and **2**
are the radical cross-coupling product of the αS radical and ketyl radical 3CBH$^{\bullet}$ after
the dehydration reaction occurring in the MS source [27–32]. We previously published a
detailed explanation of the presence of multiple isomeric αS-3CBH photoproducts in the
methionine-containing peptides and a thorough description of the diagnostic ions (circled
in red in Figure 4) [22,23]. The MSMS spectra of photoproducts **1** and **2** are shown in
Figure 4.

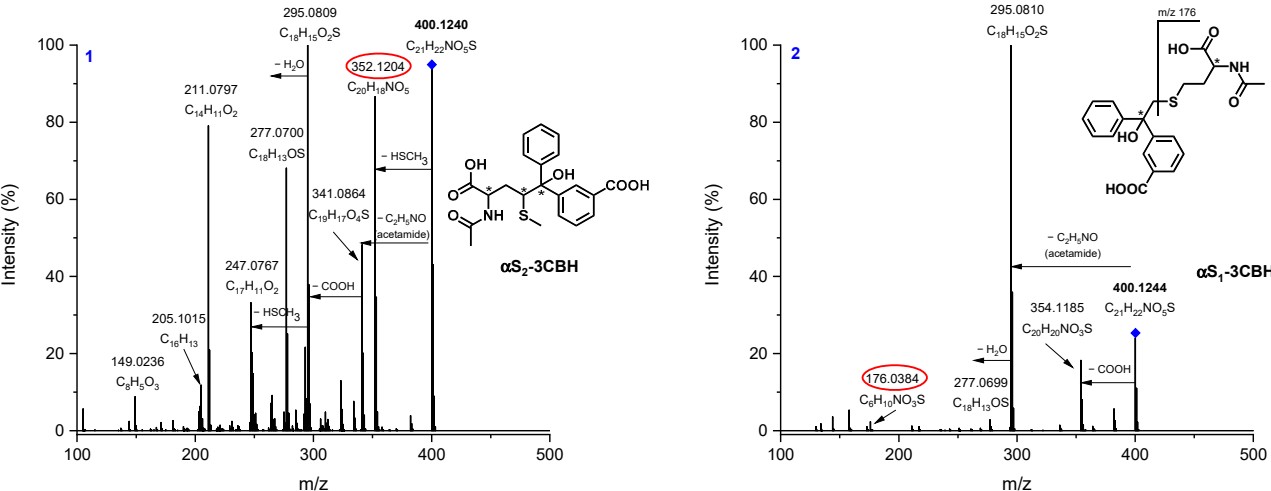

**Figure 4.** MSMS spectra of photoproducts **1a** and **1b** (fragmentation of *m/z* 400 ion).

The chromatogram, presented in Figure 3, also shows two peaks, marked **3**. Their retention times differ significantly but the accurate masses (*m/z* 356.1326), as well as the fragmentation patterns, are identical. This m/z corresponds to the molecular formula of $C_{20}H_{22}NO_3$, which can be assigned to the product of the cross-coupling reaction of the αN radical (α-amidoalkyl radical) and the 3CBH• product loss of water molecule (typical reaction occurring in MS ion source [27–32]).

αN radical is formed in the decarboxylation reaction (via pseudo-Kolbe mechanism) from sulfur-centered radical cation >S•+ [25] (Figure S3). αN-3CBH has two stereocenters: one is fixed (S) and the other one can have either a R or S configuration, giving two possible diastereoisomers (SS and SR); consequently, there are two peaks on the chromatogram corresponding to these stereoisomers.

The main fragmentation patterns of photoproducts **3** (Figures 5 and S4) correspond to the loss of the -HSCH₃ group and the amide bond cleavage (loss of the acetyl group). Other fragment ions were formed after the neutral losses of water molecules and ammonia.

The exact masses of the diagnostic ions from the MSMS experiments that allowed us to suggest the structures of photoproducts are collected in Table 2.

It is noteworthy that no products of αS-αS, αN-αN, or αS-αN radical-coupling were detected, suggesting that virtually all of the αS and αN radicals were ultimately trapped by 3CBH•.

**Table 2.** High-resolution MSMS data for the products of N-Ac-Met oxidation.

| Photoproduct | Accurate Mass (Measured) | Exact Mass (Calculated) | Mass Accuracy (ppm) | Molecular Composition |
|---|---|---|---|---|
| **1** (αS₂-3CBH) | 400.1240 | 400.1219 | 5.33 | $C_{21}H_{22}NO_5S$ |
| **2** (αS₁-3CBH) | 400.1244 | 400.1219 | 6.33 | $C_{21}H_{22}NO_5S$ |
| **3** (αN-3CBH) | 356.1326 | 356.1320 | 1.57 | $C_{20}H_{22}NO_3S$ |

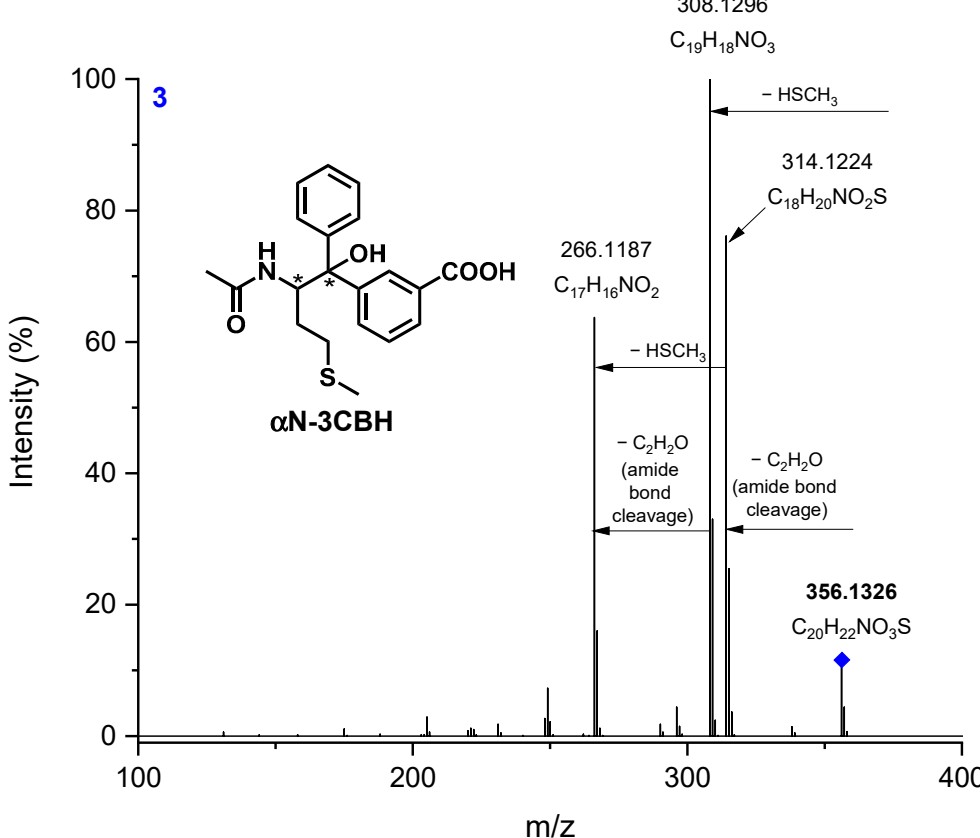

**Figure 5.** MSMS spectrum of photoproduct **3** (fragmentation of $m/z$ 356 ion).

The quantum yields of the two competing reactions, the deprotonation (formation of αS-3CBH products) and decarboxylation (formation of αN-3CBH products) of N-Ac-Met, are summarized in Table 3. The calculations were based on the quantum yield of the charge separation reaction ($k_{sep}$) and the ratio of the area under the peaks of the formed photoproducts observed on the chromatogram (Figures 3 and S1), assuming that the αS-3CBH and αN-3CBH photoproducts have the same molar absorption coefficient. The ratio of the photoproducts was the same after 10 and 30 min of irradiation.

**Table 3.** Quantum yields of two competing reactions of sulfur radical cation of N-Ac-Met (**1**) [a].

| Reaction Pathway | Φ at pH = 6.4 | Φ at pH = 10.7 |
|---|---|---|
| Deprotonation (αS) | 0.23 | 0.23 |
| Decarboxylation (αN) | 0.09 | 0.05 |

[a] ±15% experimental error.

## 4. Discussion

The suggested fate of the sulfur radical cation >S•+ is presented in Scheme 1. This cation can undergo two competing reactions: deprotonation, yielding α-thioalkyl radical (αS); or decarboxylation via a pseudo-Kolbe reaction, which involves the intramolecular electron transfer from the carboxylate group to the sulfur radical cation, leading to the decarboxylation and formation of α-amidoalkyl radical (αN).

**Scheme 1.** Two pathways of N-Ac-Met sulfur radical cation: deprotonation yielding two isomeric αS radicals, and decarboxylation yielding αN radical.

Neither αS nor αN radicals could be detected in the LFP experiments; therefore, the reaction paths of the sulfur radical cation were deduced from the analysis of the stable products, carried out using high-resolution mass spectrometry. The stable products formed in the radical cross-coupling reaction of αS and αN with 3CB ketyl are unique and sensitive markers for deprotonation and decarboxylation processes. The αN-3CBH photoproducts were formed at both pH = 6.7 and 10.7; as the carboxylic group is deprotonated (p$K_a$ of the carboxylic group of methionine is 2.16 [33]), the decarboxylation via pseudo-Kolbe mechanism is possible.

One may expect that the decarboxylation of N-Ac-Met with its free carboxylic group should be a common reaction but, contrary to our results, the previously reported quantum yields of decarboxylation in sensitized oxidation was found to be negligible [10,19,34,35]. The earlier results were based on the less sensitive techniques, e.g., laser flash photolysis, pulse radiolysis studies, and $CO_2$ detection using gas chromatography [24,36,37].

## 5. Conclusions

In the current paper, we investigated the competition between the deprotonation and decarboxylation of the sulfur radical cation >S$^{\bullet+}$ of N-Acetyl-Methionine (N-Ac-Met). All of the stable products of 3CB-sensitized oxidation of N-Ac-Met were characterized using LC-MS and MSMS. The accurate masses of the products and the MSMS fragmentation patterns provided the molecular formula and structural information of the photoproducts and, based on that information, their precursor radicals.

The α-thioalkyl radicals (αS) are formed as a result of the deprotonation of >S$^{\bullet+}$, and eventually lead to the radical cross-coupling isomeric products **1** and **2**. The α-amidoalkyl radicals (αN), resulting from the decarboxylation of >S$^{\bullet+}$, formed photoproducts **3**. We provided experimental proof for the decarboxylation of N-Ac-Met, which was previously neglected. Based on the LC-MSMS technique, we found the marker of decarboxylation with a much higher sensitivity than the previously utilized methods: the formation of the stable photoproducts αN-3CBH could act as a probe for decarboxylation as αN radicals are formed only after the loss of $CO_2$.

The calculated free-radical reaction yields showed that the preferred reaction pathway of $>S^{\bullet+}$ is deprotonation ($\phi = 0.23$ for both pH for deprotonation vs. $\phi = 0.09$ at pH 6.7 and $\phi = 0.09$ at pH 10.7 for decarboxylation).

Based on the results from the time-resolved experiments (laser flash photolysis) and stable product analysis (LC-MS and MSMS), the complete mechanism, including the quantum yields of the reactions, of the photosensitized oxidation of N-Acetyl-Methionine is proposed in Scheme 2.

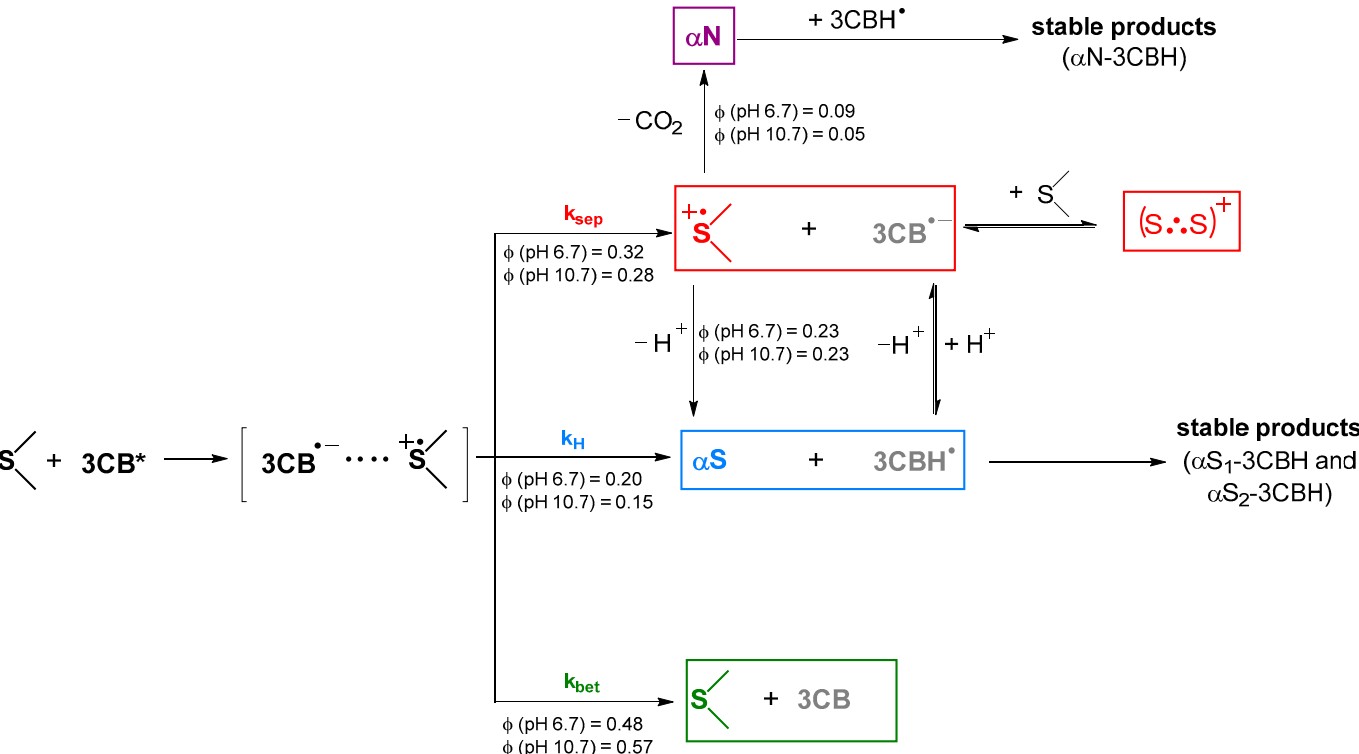

**Scheme 2.** Mechanism of 3CB* photosensitized oxidation of N-Ac-Met.

In summary, we have presented key insights into the free radical reaction pathways of the biologically relevant methionine residue, which mimics the Met35 in the C-terminal domain of β amyloid. The results of this research offer the first step in the long journey of explaining the pathway leading to the irreversible modifications and damages in protein that cause Alzheimer's disease.

**Supplementary Materials:** The following supporting information can be downloaded at: https://www.mdpi.com/article/10.3390/photochem3010007/s1, Figure S1: HPLC chromatograms of aqueous solutions containing: N-Ac-Met (20 mM) and 3CB (4.5 mM) at pH 6.7 before irradiation (top panel, red), after 10-minute irradiation (middle panel, green) and after 30-minute irradiation (bottom panel, blue). The expansion of the chromatograms after irradiation between 12 and 18 minutes is shown next to the chromatograms. Peaks are labelled with m/z of the [M + H]+ or [M + H-H$_2$O]+ ions of the acquired MS spectrum.; Figure S2: HPLC chromatograms of aqueous solutions containing: N-Ac-Met (20 mM) and 3CB (4.5 mM) at pH 10.7 before irradiation (top panel, orange), after 10-minute irradiation (middle panel, dark green) and after 30-minute irradiation (bottom panel, purple). The expansion of the chromatograms after irradiation between 12 and 18 minutes is shown next to the chromatograms. Peaks are labelled with m/z of the [M + H]$^+$ or [M + H-H$_2$O]$^+$ ions of the acquired MS spectrum.; Figure S3: The mechanism of the pseudo-Kolbe reaction.; Figure S4: MSMS spectra of photoproducts 3 at retention time 14.5 and 17.3 min.

**Author Contributions:** Conceptualization, T.P.; methodology, T.P., V.S. and K.G.; validation, T.P.; investigation, K.G. and V.S.; writing—original draft preparation, V.S, K.G.; writing—review and editing, T.P. and K.G.; visualization, V.S., K.G.; supervision, T.P.; funding acquisition, T.P., K.G. All authors have read and agreed to the published version of the manuscript.

**Funding:** This research was supported by the Initiative of Excellence—Research University at Adam Mickiewicz University, project no. 006/07/POB3/0004 and project no. 017/02/SNŚ/0008.

**Data Availability Statement:** Not applicable.

**Acknowledgments:** Not applicable.

**Conflicts of Interest:** The authors declare no conflict of interest.

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
