# Peer review of "The Fate of Sulfur Radical Cation of N-Acetyl-Methionine: Deprotonation vs. Decarboxylation"

_2673-7256, doi:10.3390/photochem3010007_

Round 1

Reviewer 1 Report

The manuscript by Grzyb et al. presents a LCMS and MSMS study on the fate of the (N-acetylated) methionine sulfur radical cation, which is generated through photooxidation using 3-carboxybenzophenone as sensitizer in aqueous solution at neutral and alkaline pH. 

There are several reactions for the S-radical cation, one is formation of an 2c3e S-dimer, whereas other pathways involve deprotonation or decarboxylation following an intramolecular ET. 

The radical species were identified as the closed-shell adducts with the photoproduct of the sensitizer through LCMS and HPLC. The major pathway is formation of the two isomeric alpha-S radicals, whereas decarboxylation is a minor (but not negligible )pathway.

The experiments have been performed well, as far as I can judge, and the manuscript is worthy to be published.

I have only a few minor comments/queries:

a) abstract - 'deprotonation was found to be favored over deprotonation"......

b)  Figure 2 - why is the concentration of transients lower in the alkaline environment than in the neutral? Likewise, why do the blue and red transients decay slower in alkaline solution than in neutral?

c) Figure 3 - these are LCMS plots, right? It would be useful if the caption states that the small numbers in the chromatograms are m/z values.

d) Figure 4 - why are certain masses circled in red? 

e) Figure 5 - fragmentation of m/z 314 to m/z 266 through loss of CH3SH and not starting from m/z 308. 

Author Response

Thank you for your valuable comments and suggestions. They are very helpful for revising and improving our paper. We have studied comments carefully and have made correction which we hope meet your approval (revised portions are marked in red in the revised manuscript):

a) corrected

b) There are mostly two transients decaying in Figure 2: SS+ and 3CB(dot)-. The apparent slower decay of transients at basic pH (Fig. 2D) is associated with a different time-scales when Fig. 2B is compared to Fig. 2D. It's tru that 3CB radical anion decays faster in neutral solution (pH=6.7). The faster decay is certainly associated with its protonation (eventually leading to 3CBH(dot)).

c) Figure 3 caption has been corrected according to the reviewer's suggestion

d) the m/z values circled in red are so called diagnostic ions. In other words - ions characteristic for given fragmentation pattern, proving the molecular structure - explanation has been added to the text in line 179

e) Figure 5 has been corrected according to the reviewer's suggestion

Reviewer 2 Report

The authors of this paper investigated the photooxidation of N-acetyl methionine using 3-carboxybenzophenone as a sensitizer. They combined a time resolved identification of transient species with laser flash photolysis and final product analysis with mass spectrometry coupled to chromatography. This study is well-conducted and gives interesting results. I just regret that the authors didn’t compare their results with what can be obtain with normal methionine. This would justify the choice of only one protecting group and its position, as I am not really convinced with the argument of mimicking the b-amyloid oxidation (see below).

Comments:

Abstract m13/14. There must be an error in the sentence with twice the word deprotonation, as we can read:”the deprotonation was found to be favored upon deprotonation”.

Abstract: they author justify the choice of N-Ac-Met as it should mimic the methionine in the C-terminal domain of b-amyloid. I don’t totally agree with this argument because in b-amyloid, the methionine residue is not terminal so it shouldn’t possess and COOH function. Or is there a reason for just protecting the amino part of the residue?

Introduction l21: the authors mention “the resulting modifications” but it is not clear for me which modifications and induced by which phenomenon.

L151: How were quantified initial concentrations of 3CB*? Did the authors used e values? And if so, what are these values? Because I expect these values to be different for different pH values.

All over the text (example l166, 167) and in all figures, I would rather mentioned exact masses.

According to the authors, they were able to detect all stable products of 3CB sensitized oxidation of N-Ac-Met. From Figure 3, many chromatographic peaks can be clearly seen between 15 and 17 minutes that seem to increase with the irradiation time (supplementary). What are the masses of these species and did the authors identified them?

Author Response

Thank you for your valuable comments and suggestions. They are very helpful for revising and improving our paper. We have studied comments carefully and have made correction which we hope meet your approval (revised portions are marked in red in the revised manuscript):

  • "Abstract m13/14. There must be an error in the sentence with twice the word deprotonation, as we can read:”the deprotonation was found to be favored upon deprotonation” - corrected
  • "Abstract: they author justify the choice of N-Ac-Met as it should mimic the methionine in the C-terminal domain of b-amyloid. I don’t totally agree with this argument because in b-amyloid, the methionine residue is not terminal so it shouldn’t possess and COOH function. Or is there a reason for just protecting the amino part of the residue?" - the reviewer is absolutely right - Met is NOT a terminal aminoacid in beta-amyloid peptide, however, the model compound of choice (NAc-Met) with its free carboxylic group allows decarboxylation studies. The stable products of decarboxylation (detected with MS and MS/MS) can then be used as markers to investigate this reaction in more complex systems (e.g. beta-amyloid). That was our only intention when saying in that NAc-Met can "mimic" more complex peptides. The abstract has been re-phrased accordingly.
  • "Introduction l21: the authors mention “the resulting modifications” but it is not clear for me which modifications and induced by which phenomenon" - the sentence has been re-phrased according to the reviewer's suggestion
  • "L151: How were quantified initial concentrations of 3CB*? Did the authors used e values? And if so, what are these values? Because I expect these values to be different for different pH values." - the 3CB triplet concentrations were calculated from Laser Flash Photolysis experiments using relative actinometry method (with known e values) - they are 3.46 uM and 1.70 uM for neutral and alkaline solutions, respectively (see line 152-153)
  • "All over the text (example l166, 167) and in all figures, I would rather mentioned exact masses" - the m/z values in the text have been changed to exact masses according the reviewer's suggestion. However we believe the m/z values labeling the chromatographic peaks (e.g. Figure 3) should stay as they are (for better, more clear reading)
  • "According to the authors, they were able to detect all stable products of 3CB sensitized oxidation of N-Ac-Met. From Figure 3, many chromatographic peaks can be clearly seen between 15 and 17 minutes that seem to increase with the irradiation time (supplementary). What are the masses of these species and did the authors identified them?" - correct, there are a few small, unidentified peaks on the chromatogram, however, we set a certain threshold for identification and decided "not to worry" about peaks below that line. The small peaks the reviewer refers to are most likely due to secondary reactions.